# SEARCH-ADAPTOR: TEXT EMBEDDING CUSTOMIZATION FOR INFORMATION RETRIEVAL

## ABSTRACT

Text embeddings extracted by pre-trained Large Language Models (LLMs) have significant potential to improve information retrieval and search. Beyond the zero-shot setup in which they are being conventionally used, being able to take advantage of the information from the relevant query-corpus paired data has the power to further boost the LLM capabilities. In this paper, we propose a novel method, Search-Adaptor, for customizing LLMs for information retrieval in an efficient and robust way. Search-Adaptor modifies the original text embedding generated by pre-trained LLMs, and can be integrated with any LLM, including those only available via APIs. On multiple English and multilingual retrieval datasets, we show consistent and significant performance benefits for Search-Adaptor – e.g., more than 5.2% improvements over the Google Embedding APIs in nDCG@10 averaged over 14 BEIR datasets.

## 1 INTRODUCTION

Information retrieval can be broadly considered as the task of searching for information via querying a corpus database that might consist many different types of data, such as documents, webpages or logs. There are wide range of applications across many industries, including retail, healthcare, and finance, with a significant portion of the world's economy is built on. Particularly, language modeling is the key part of information retrieval as in most cases, query and corpus data are in text form. Large language models (LLMs) have demonstrated significant achievements for a variety of text processing tasks, including question answering, summarization, and mathematical reasoning (1; 2; 3). One critical enabler for the success on these has been transforming raw text into meaningful representations that preserve semantic meanings in the representation space (4). For a wide range of applications, from recommendations to anomaly detection, tasks are defined as explicit operations on the learned representations. This makes the quality of the text mapping into embeddings become of paramount importance. For information retrieval, a common approach has been to directly utilize the text embeddings, where relevant corpora can be ranked based on the similarity between queries and corpus (5; 6).

Various LLMs have been proposed to extract embeddings from raw text, including the Sentence T5 (7), OpenAI embedding APIs (8) and Google embedding APIs (9). However, one fundamental limitation of pre-trained LLMs is that they cannot utilize paired samples in the form of (positive query-corpus pairs), if they exist. Even with a small number of paired samples, those samples are expected to significantly improve information retrieval capabilities. In this paper, we focus on customizing LLMs to obtain superior embeddings for information retrieval applications using those paired samples. Fig. 1 overviews text embedding customization for superior retrieval.

Full fine-tuning (10) can be the straightforward way of utilizing the paired query-corpus information. However, if the number of paired samples is small, tuning all the weights of a model might yield overfitting and poor generalization (11), especially in the presence of distribution shifts. In addition, full fine-tuning can be costly from a computational perspective as it requires large memory. There are multiple parameter-efficient fine-tuning methods such as prompt tuning (12; 13), LoRA (14), partial fine-tuning (15), and various adapters (16; 17). Those models only fine-tune a subset of the parameters in LLMs, which reduces the risks of overfitting and provides computational gains. As a common bottleneck, all of these methods need full access to the LLM's parameters to fine-tune the model, which may not be possible with API-based LLMs.

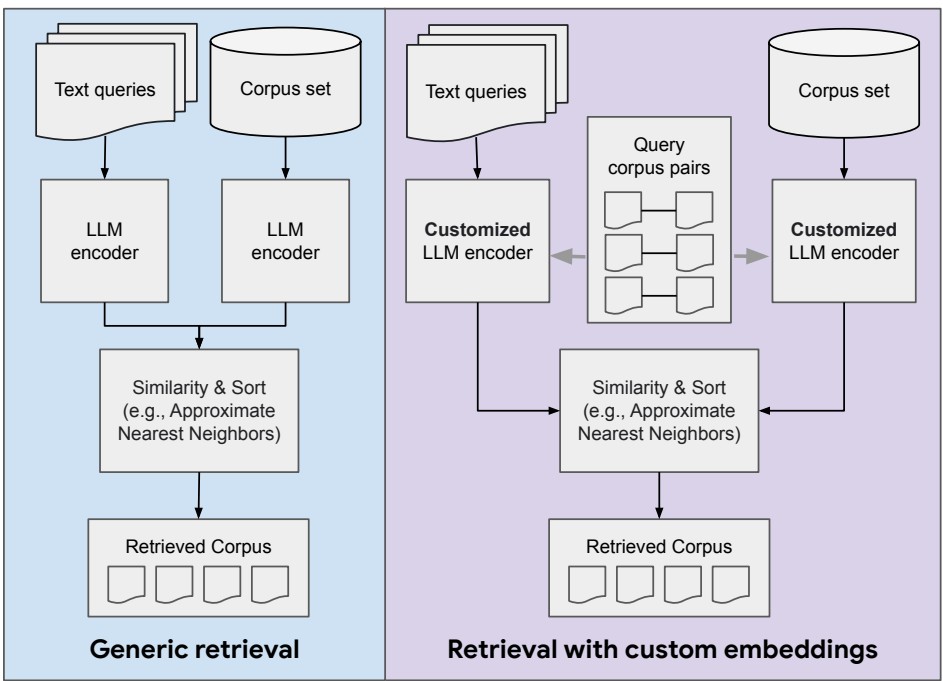

Figure 1: Comparison between generic retrieval (blue) and retrieval with embedding customization (purple). Search-Adaptor modifies the pre-trained LLM embeddings (using positive query-corpus pairs) so that they are customized on the given dataset. Note that Search-Adaptor does not require full access to the LLM parameters.

We propose a novel adaptation approach, Search-Adaptor, that places a small adapter network (customized for the given dataset) on top of fixed LLMs to modify the pre-trained text embeddings. We introduce a novel differentiable ranking loss that can directly utilize the information of positive query and corpus pairs. In addition, we include multiple regularizers to improve generalization in this small data regime where without intervention the pre-trained LLMs would lead to catastrophic forgetting. One major advantage of Search-Adaptor is that it does not require access to the parameters of the pre-trained LLMs – only the inference outputs of the model are needed. Commercial embedding APIs that show state-of-the-art performance usually do not provide access to their model parameters. In such cases, Search-Adaptor can still be used to further improve those API-based embedding models, in contrast to alternative methods such as full fine-tuning and parameter-efficient fine-tuning. We demonstrate the efficacy of Search-Adaptor across 14 BEIR datasets (18) and 17 multilingual datasets (MIRACL (19)) with Google and OpenAI embedding APIs, applying the Search-Adaptor on top. In addition, we evaluate Search-Adaptor's performance improvements with 2 different Sentence T5 models (7) that provide full access to their parameters. Overall, Search-Adaptor provides consistent and significant improvements over alternatives. The contributions of this paper can be summarized as follows.

- We propose a novel adaptation framework for information retrieval applications that can significantly improve the pre-trained large language embedding models.
- We introduce a novel ranking loss and multiple regularizers that reduce overfitting and forgetting and thereby improve the retrieval performance even with the small data regime.
- We provide consistent and significant improvements for retrieval performance with a range of datasets (from 1K to 500K positive query-corpus training data pairs).

## 2 RELATED WORKS

**Pre-trained LLMs for zero-shot retrieval.** LLMs to extract general text embeddings are commonly used in both academia and industry. AI solution providers like Google (9) and OpenAI have

productionized general text embeddings that can be directly used via simple APIs for zero-shot retrieval applications. In addition, multiple previous works introduce new general text embedding models with various pre-training methods and datasets. GTE (20) proposes a multi-stage pre-training of embedding models with diverse naturally paired text datasets. E5 (5) pre-trains the embedding models by weakly-supervised contrastive learning, utilizing consistency-based filter to generate high quality text pairs for pre-training. Note that Search-Adaptor can be applicable on top of any pre-trained LLM embedders to customize their embeddings for superior retrieval performances.

**Embedding customization.** Instead of using one unified model for zero-shot retrieval, the embeddings can be customized for each dataset or task. Instruction-based embedding customization is one popular method. TART (21) builds a retrieval system that adapts the retrieval based on the instruction. Different retrieval tasks (e.g., code, question, or answer) are given as the instruction to further improve dense embedding retrieval. InstructOR (22) integrates the task and domain descriptions prior to the input to fine-tune the embeddings for retrieval. However, these do not directly utilize the provided positive query and corpus pairs. Full or parameter-efficient fine-tuning (such as LoRA (14) and (IA)$^3$ (23)) can also be considered for embedding customization. Pre-trained LLMs can be fine-tuned with contrastive loss using positive query-corpus paired data. Promptagator (24) utilizes in-context learning to generate synthetic query-corpus pairs using a few number of original query-corpus pairs, and subsequently using those synthetic pairs to fine-tune the pre-trained LLMs. However, all these are only applicable when there is full access to the parameters of pre-trained LLMs, which is often not possible for state-of-the-art commercial text embedding models. On the other hand, Search-Adaptor can be applied without full access to the LLM parameters.

## 3 PROBLEM FORMULATION

We formulate the retrieval problem with a given query-corpus paired dataset. Assume a query set denoted as $\mathcal{Q} = \{q_1, ..., q_N\} \in Q$ and a corpus set denoted as $\mathcal{C} = \{c_1, ..., c_M\} \in C$. Each positive relationship between a query and corpus is represented as the triplet $r_{ij} = (q_i, c_j, y_{ij})$ with $y_{ij} > 0$ as the strength of the relationship between $q_i$ and $c_j$. We treat all other triplets as negative relationships ($y_{ij} = 0$). The set of all query-corpus relationships is denoted as $\mathcal{R} = \{(q_i, c_j, y_{ij})\}_{i=1:N, j=1:M} = \mathcal{R}_p \cup \mathcal{R}_n$, where $\mathcal{R}_p = \{(q_i, c_j, y_{ij}) \in \mathcal{R} | y_{ij} > 0\}$ is the set of positive relationships and $\mathcal{R}_n = \{(q_i, c_j, y_{ij}) \in \mathcal{R} | y_{ij} = 0\}$ is the set of negative relationships. Note that $y_{ij}$ can either be binary or continuous.

The retrieval system aims to find the relationship between the given query ($q_i$) and corpus ($c_j$) such that the predicted relationship is highly correlated with the ground truth relationship ($y_{ij}$). The scoring function $f : Q \times C \to \mathbb{R}$ takes queries and corpus data as inputs and outputs a score estimate on the relationship between them. The optimal score is the one that has the same order as the ground truth relationship for each query.

## 4 METHODS: SEARCH-ADAPTOR

We next describe our proposed method, Search-Adaptor, to customize embeddings extracted from the pre-trained LLMs. In the following subsections, we introduce the ranking loss that is directly utilized for Search-Adaptor training. Then, we present two regularizers that help avoid overfitting on training data. Fig. 2 describes the entire block diagrams of Search-Adaptor.

### 4.1 ADAPTING FIXED LLMS

Real-world limitations arise when tuning the embedding model: it can be very costly and one often does not have access to the parameters and the gradients of the pre-trained model (for example, it may only be available via an inference API). This motivates the need for an adaptation method that can operate with fixed pre-trained embedding models, such as the ones that are only accessible via APIs for inference via prompting. To this end, we propose Search-Adaptor, which modifies embeddings extracted from pre-trained LLMs for superior search and information retrieval.

Consider the query and corpus embeddings extracted using the pre-trained embedding model $E$: $\mathcal{Q}_E = \{qe_1, ..., qe_N\} \in \mathbb{R}^d$ and $\mathcal{C}_E = \{ce_1, ..., ce_N\} \in \mathbb{R}^d$ where $qe_i = E(q_i)$ and $ce_j = E(c_j)$. Note that both query and corpus embeddings are in the same embedding space.

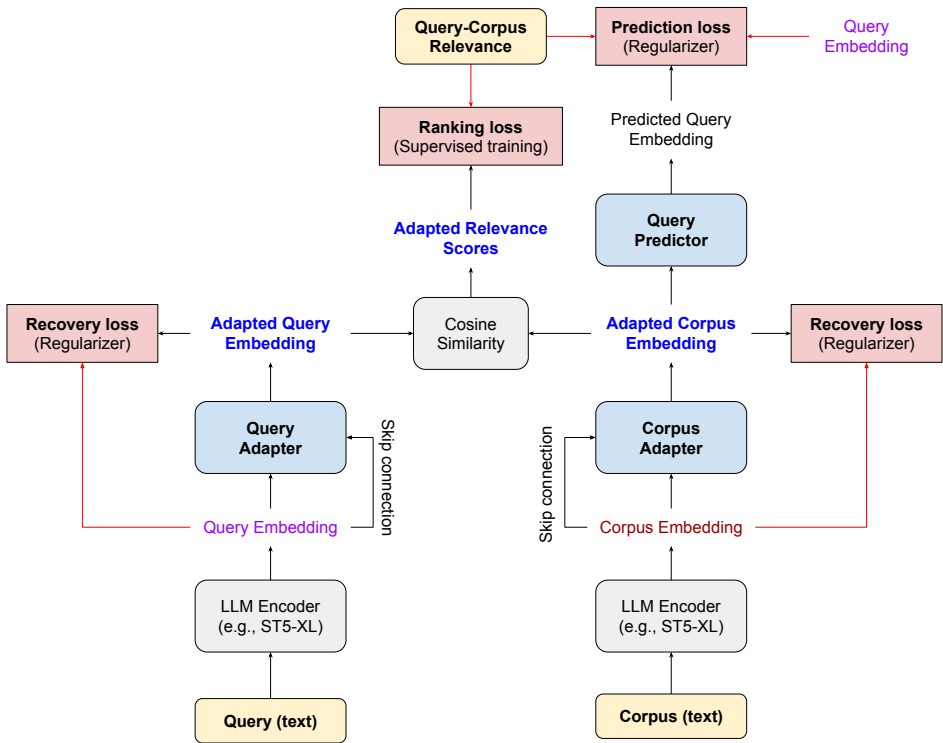

Figure 2: Block diagram of Search-Adaptor. Yellow-colored blocks are given inputs; grey colored blocks are fixed LLM encoders (e.g., a text embedding API); blue-colored blocks are additional trainable building blocks; red-colored blocks are for loss computations. At inference, only query and corpus adapters are utilized and the query predictor can be discarded.

The objective of Search-Adaptor is to modify embeddings extracted from pre-trained LLMs in a way that maximizes retrieval performance. The learnable adaptation function is defined as $f : \mathbb{R}^d \to \mathbb{R}^d$, which maps the original embedding to a new embedding that is more favorable for retrieval applications. The modified embeddings are denoted as $\hat{\mathcal{Q}}_E = \{\hat{qe}_1, ..., \hat{qe}_N\} \in \mathbb{R}^d$ and $\hat{\mathcal{C}}_E = \{\hat{ce}_1, ..., \hat{ce}_M\} \in \mathbb{R}^d$ where $\hat{qe}_i = f(qe_i)$ and $\hat{ce}_j = f(ce_j)$. The relevance scores between modified query and corpus embeddings are defined as follows:

$$\hat{s}_{ij} = \text{Cosine-Similarity}(\hat{qe}_i, \hat{ce}_j) = \frac{\hat{qe}_i \cdot \hat{ce}_j}{||\hat{qe}_i||||\hat{ce}_j||}.$$

Search-Adaptor consists of the following components (see Fig. 2 for details):

- **Adaptation function** $f$. This function is used to modify the query and corpus embeddings. We add a skip connection to $f$ so that it can only learn the residual between the original and adapted embeddings as follows: $\hat{qe}_i = qe_i + f(qe_i)$ and $\hat{ce}_j = ce_i + f(ce_i)$. Note that we use the shared adapter for both query and corpus (see Sec. 6 for ablation studies). The ranking loss, reconstruction loss, and prediction loss are used to train $f$.
- **Query predictor** $p$. This function is used to predict the query embedding using the adapted corpus embedding. The prediction loss is used to train $p$.

At inference, we only use the adaptation functions ($f$) to modify the query and corpus embeddings. We then compute the cosine similarity between the modified query and corpus embeddings to estimate the relevance between query and corpus. Query predictor is not used at inference.

## 4.2 RANKING OBJECTIVE

As explained in Sec. 3, the objective of the retrieval is to predict the correct order of the relevance between queries and corpus. Therefore, the most critical part is to properly design the ranking loss.

We propose a ranking loss as follows:

$$\mathcal{L}_{Rank} = \sum_{i=1}^{N} \sum_{j=1}^{M} \sum_{k=1}^{M} I(y_{ij} > y_{ik}) \times (y_{ij} - y_{ik}) \times \log(1 + e^{(s_{ik} - s_{ij})}),$$

where $I(y_{ij} > y_{ik})$ is an indicator function that is equal to 1 if $y_{ij} > y_{ik}$ and 0 otherwise. The pre-trained embedding model is denoted as $E$, which converts raw text into embeddings. The relevance score between query text ($q_i$) and corpus text ($c_j$) is defined as $s_{ij} = $ Cosine-Similarity($E(q_i), E(c_j)$).

The ranking loss penalizes the model more when it predicts a lower score for a pair of query and corpus that has a higher ground truth relevance (i.e., $s_{ij} < s_{ik}$ even though $y_{ij} > y_{ik}$). The amount of penalization is proportional to the difference in ground truth relevance ($y_{ij} - y_{ik}$) and the difference in estimated scores $\log(1 + e^{(s_{ik} - s_{ij})})$. Note that $\log(1 + e^{(s_{ik} - s_{ij})})$ can be replaced with any monotonic function such as linear function. In general, the ranking loss encourages the model to predict higher scores for pairs of query and corpus that have a higher ground truth relevance. Table 5 shows a comparison of this ranking loss to alternatives and demonstrates its effectiveness.

### 4.3 REGULARIZATION

Introducing proper inductive biases via regularization is important to improve adaptation from pre-trained LLM embeddings without forgetting too much information from the pre-trained LLMs. Towards this end, we propose two regularization methods:

**Recovery.** To increase generalizability, we postulate avoiding modification of the adapted embedding too far away from the original embedding. As such, we minimize the difference between the original and adapted embeddings using a recovery regularizer, which is calculated as follows:

$$\mathcal{L}_{Recovery} = \frac{1}{N} \sum_{i=1}^{N} ||\hat{q}e_i - qe_i||_1 + \frac{1}{M} \sum_{j=1}^{M} ||\hat{c}e_i - ce_i||_1$$

where $\hat{q}e_i$ is the adapted query embedding and $qe_i$ is the original query embedding. Similarly, $\hat{c}e_i$ is the adapted corpus embedding and $ce_i$ is the original corpus embedding. The recovery regularizer encourages the adapted embeddings to be not too far from the original embeddings.

**Prediction.** Intuitively, if the query and corpus are highly relevant, we can use the corpus to predict the query. Building upon this intuition, we propose a regularizer in the form of prediction loss between the query and corpus, calculated as follows:

$$\mathcal{L}_{Pred} = \frac{1}{\sum_{i=1}^{N} \sum_{j=1}^{M} y_{ij}} \sum_{i=1}^{N} \sum_{j=1}^{M} y_{ij} \times ||\hat{q}e_i - p(\hat{c}e_j)||_1$$

where $p : \mathbb{R}^d \rightarrow \mathbb{R}^d$ is a function that predicts the query given the corpus, and $y_{ij}$ is a weight that is assigned to the loss if the query and corpus are correlated. The prediction loss encourages the model to predict the query well given the corpus, especially if the query and corpus are correlated. Note that we do not include the prediction function from query to corpus because usually corpora are significantly longer than queries which would render the task challenging.

### 4.4 TRAINING

Using the proposed ranking loss, recovery loss, and prediction loss, we optimize the adaptation function $f$ and prediction function $p$ by minimizing the following loss function:

$$f^*, p^* = \arg\min_{f,p} \mathcal{L}_{Rank}(f) + \alpha \mathcal{L}_{Recovery}(f) + \beta \mathcal{L}_{Pred}(f, p),$$

where $\alpha \geq 0$ and $\beta \geq 0$ are hyper-parameters that control the relative importance of the different loss terms. In the experiments, we tune these hyper-parameters based on validation set ($\alpha \in \{0.0, 0.1, 1.0\}$ and $\beta \in \{0.0, 0.01, 0.1\}$). Table 5 shows the results of ablation studies on the effectiveness of the different loss terms. All hyper-parameters are provided in Appendix Sec. C.

Note that the ranking loss compares all possible pairs between queries and corpus which needs $NM^2$ times computations per one epoch ($M >> N$). For efficient computation, we randomly subsample the corpus for each query batch. While doing so, we always include the corpus which has positive relevance to queries in that batch.

## 5 EXPERIMENTS

We evaluate the performance of Search-Adaptor in multiple scenarios on numerous datasets. We demonstrate that Search-Adaptor is model-agnostic, applying it both on top of API-based LLMs (merely via access to Google & OpenAI APIs) and open-sourced LLMs (Sentence T5 models (7)). We also demonstrate that it is data-agnostic by evaluating Search-Adaptor on both English and non-English multilingual datasets.

### 5.1 EXPERIMENTAL SETTINGS

We first consider the 14 retrieval datasets from the BEIR repository (25) to evaluate the performance in English data, with corpus sizes ranging from 3.6K to 8.8M, and training pairs ranging from 0.7K to 532K. For the datasets with only test data (e.g., Arguana, SciDocs), we split the data into disjoint train and test sets with a 50/50 ratio, based on the sorted query IDs. We also use MIRACL (26) which consists of 17 multilingual datasets including Japanese, Chinese, French, and Indonesian. More details can be found in Appendix A.

We use nDCG@10 as the main metric for the retrieval performance (see Appendix B for more details). For model selection, we further divide the training data into disjoint training and validation datasets with an 80/20 ratio and select the model with the highest validation nDCG@10 metric.

We consider both API-based and open-sourced LLMs. As the API-based LLM, we use OpenAI embedding API (8) and Google embedding API (9). As the open-sourced LLM, we use Sentence T5 models[1] of two different sizes.

### 5.2 ADAPTING WITH API-BASED LLMS

| Datasets | Google Embedding API | | | OpenAI Embedding API | | |
|---|---|---|---|---|---|---|
| | Zero-shot | Search-Adaptor | Gains (%) | Zero-shot | Search-Adaptor | Gains (%) |
| NFCorpus | 0.3794 | **0.3829** | 0.92% | 0.3750 | **0.3785** | 0.93% |
| SciFact | 0.7075 | **0.7931** | 12.10% | 0.7221 | **0.7904** | 9.46% |
| Arguana | 0.5308 | **0.7047** | 32.76% | 0.5885 | **0.6493** | 10.33% |
| SciDocs | 0.1887 | **0.2135** | 13.14% | 0.2003 | **0.2158** | 7.74% |
| FiQA | 0.4901 | **0.5224** | 6.59% | 0.4366 | **0.4410** | 1.01% |
| Trec-Covid | 0.7278 | **0.7530** | 3.46% | 0.7224 | **0.7733** | 7.05% |
| Webis Touche 2020 | 0.2491 | **0.3339** | 34.04% | 0.2590 | **0.3312** | 27.88% |
| Quora | 0.8614 | **0.8765** | 1.75% | 0.8830 | **0.8869** | 0.44% |
| NQ | 0.5205 | **0.5485** | 5.38% | - | - | |
| DBPedia | 0.4038 | **0.4051** | 0.32% | - | - | |
| HotPotQA | 0.6443 | **0.6839** | 6.15% | - | - | |
| Fever | 0.8252 | **0.8566** | 3.81% | - | - | |
| Climate-fever | 0.2272 | **0.3204** | 41.02% | - | - | |
| MSMARCO | 0.2922 | **0.3177** | 8.73% | - | - | |

Table 1: Performance improvements with Search-Adaptor for two API-based LLMs. The embedding dimensions of Google (gecko@latest) and OpenAI APIs (text-embedding-ada-002) are 768 and 1536, respectively.

One of the biggest advantages of Search-Adaptor is that it can be applied on top of any API-based LLM – without having access to the parameters of LLMs, Search-Adaptor can further improve

---

[1] https://tfhub.dev/google/sentence-t5/st5-base/1

the retrieval performance. This is particularly important as the state-of-the-art LLMs are actually API-based models (owned by companies).

As can be seen in Table 1, we demonstrate the retrieval performance improvements on top of API-based text embedding models across 14 datasets from the BEIR repository. On average, Search-Adaptor achieves 0.0475 and 0.0349 nDCG@10 improvements for both Google and OpenAI text embedding APIs. For some datasets, the improvements are quite significant indeed – e.g., 0.1739 with Arguana, 0.0856 with Scifact datasets.

| < 2M Corpus | | | | >= 2M Corpus | | | |
|---|---|---|---|---|---|---|---|
| Dataset | Zero-shot | Search-Adaptor | Gain (%) | Dataset | Zero-shot | Search-Adaptor | Gain (%) |
| Bengali | 0.6641 | **0.7141** | 7.54% | Persian | 0.5026 | **0.5229** | 4.04% |
| Hindi | 0.5250 | **0.5286** | 0.69% | Arabic | 0.6324 | **0.6809** | 7.67% |
| Swahili | 0.6717 | **0.7156** | 6.54% | Chinese | 0.4673 | **0.5242** | 12.18% |
| Telugu | 0.7407 | **0.7999** | 7.99% | Spanish | 0.4774 | **0.5031** | 5.38% |
| Thai | 0.6422 | **0.7109** | 10.70% | French | 0.3813 | **0.4286** | 12.40% |
| Yoruba | 0.7709 | **0.8506** | 10.34% | Japanese | 0.5373 | **0.5689** | 5.88% |
| Indonesian | 0.4465 | **0.5000** | 11.98% | Russian | 0.5283 | **0.5426** | 2.71% |
| Korean | 0.5593 | **0.6051** | 8.19% | Germany | 0.4809 | **0.4826** | 0.35% |
| Finnish | 0.6646 | **0.6863** | 3.27% | | | | |
| Average | 0.6317 | **0.6790** | 7.49% | Average | 0.5009 | **0.5317** | 6.15% |

Table 2: Performance improvements with Search-Adaptor on top of the Google embedding APIs (gecko-multilingual@latest) for non-English data.

Search-Adaptor is also applicable on non-English data. In Table 2, Search-Adaptor shows consistent performance improvements on top of Google Embedding API across 17 different languages (on average 0.0396 nDCG@10 improvement). For some languages, it is particularly significant, e.g. the improvement is 0.687 for Thai. These overall highlight Search-Adaptor being a model-agnostic and data-agnostic approach.

## 5.3 ADAPTING WITH OPEN-SOURCED LLMS

Beyond API-based LLMs, Search-Adaptor can also be applied to open-sourced LLMs. In this section, we use Sentence T5-Base model as the open-sourced LLM to demonstrate the performance improvements.

| Datasets | Zero-shot | Search-Adaptor | Fine-tuning |
|---|---|---|---|
| NFCorpus | 0.3100 | 0.3258 | **0.3501** |
| SciFact | 0.5237 | 0.7255 | **0.7542** |
| Arguana | 0.3646 | 0.5501 | **0.6239** |
| SciDocs | 0.1393 | **0.1657** | 0.1640 |
| FiQA | 0.4064 | 0.4416 | **0.4557** |
| Trec-Covid | 0.5990 | **0.6986** | 0.4178 |
| Webis Touche 2020 | 0.2291 | **0.3393** | 0.1844 |
| Quora | 0.7484 | **0.8664** | 0.7817 |
| Average | 0.4151 | **0.5141** | 0.4151 |

Table 3: Performance improvements with Search-Adaptor on top of open-sourced embedding model.

In Table 3, Search-Adaptor shows consistent improvements over zero-shot ST5-Base model. On average, it shows 0.1010 nDCG@10 improvements which is much larger than the improvements for with Google and OpenAI embedding APIs. With the open-sourced LLMs, we can utilize fine-tuning methods as the alternative of Search-Adaptor (even though its training cost is much higher). The

experimental results show that on average, fine-tuning performances are indeed worse than Search-Adaptor performance for the considered benchmarks. Surprisingly, the performance of fine-tuning method is much worse than the zero-shot baseline (e.g., for Trec-Covid, Webis Touche 2020) which can be attributed to overfitting and poor generalization (11).

## 6    DISCUSSIONS

### 6.1    SMALL LLMS WITH EMBEDDING CUSTOMIZATION OUTPERFORM ZERO-SHOT LARGE LLMS

LLM inference can be costly with high latency, that can constitute bottlenecks for real-world deployments. The cost and latency of LLM inference are highly dependent on the LLM model size. We demonstrate that Search-Adaptor can achieve better or comparable retrieval performances even with much smaller LLM models in comparison to zero-shot retrieval systems.

| Base LLMs | ST5-Base | | ST5-Large | |
|---|---|---|---|---|
| Datasets | Zero-shot | Search-Adaptor | Zero-shot | Search-Adaptor |
| NFCorpus | 0.3100 | **0.3258** | 0.3354 | **0.3410** |
| SciFact | 0.5237 | **0.7255** | 0.5801 | **0.7530** |
| Arguana | 0.3646 | **0.5501** | 0.2662 | **0.4770** |
| SciDocs | 0.1393 | **0.1657** | 0.1618 | **0.1850** |
| FiQA | 0.4064 | **0.4416** | 0.4785 | **0.5028** |
| Trec-covid | 0.5990 | **0.6986** | 0.6471 | **0.7082** |
| Webis Touche 2020 | 0.2291 | **0.3393** | 0.2624 | **0.3408** |
| Quora | 0.7484 | **0.8664** | 0.7560 | **0.9705** |
| Average | 0.4151 | **0.5141** | 0.4607 | **0.5223** |

Table 4: The performances of Search-Adaptors applied on top of 2 pre-trained LLM encoder: Sentence-T5 models: (i) ST5-Base (110M parameters) and (ii) ST5-Large (335M parameters) in terms of nDCG@10.

In Table 4, Search-Adaptor with ST5-Base model (110M parameters) performs much better than ST5-Large (335M parameters). Search-Adaptor can achieve better results with much smaller encoders, which can significantly decrease the serving cost and latency of retrieval systems. It also reiterates the generalizability of Search-Adaptor across different LLMs.

### 6.2    ABLATION STUDIES

Search-Adaptor proposes multiple innovations to improve the adaptation performance. In this subsection, we quantify the contributions of proposed constituents to the retrieval performance on various datasets with as ST5-Base as the base embedding model.

In Table 5, we would like to understand the source of gains in the Search-Adaptor approach. So, we make various modifications to the Search-Adaptor: (i) different architecture, (ii) different regularization, (iii) different losses. First, using different losses makes the biggest performance degradation which represents the importance of our ranking loss function. In addition, if we use separate adapters for query and corpus, it also makes a noticeable performance drop. This shows the importance of "shared embedding space" between query and corpus for the retrieval application. Lastly, skip connection, two regularization functions also bring additional performance gains but the impact is lower than our ranking losses.

More specifically on the lower part of Table 5, we shows the impact of the proposed ranking loss in comparison to alternatives (30). We compare the proposed ranking loss with four popular alternatives: (i) point-wise sigmoid cross entropy, (ii) contrastive loss (27), (iii) softmax cross entropy (28) and (iv) RankNet loss (29). As can be seen in Table 5, with the proposed ranking loss (Original Search-Adaptor), it shows significant performance improvements in comparison to the alternative ranking losses.

|  | NFCorpus | SciFact | Arguana | SciDocs | FiQA | Trec-covid |
|---|---|---|---|---|---|---|
| Zero-shot | 0.3100 | 0.5237 | 0.3646 | 0.1393 | 0.4064 | 0.5990 |
| Original Search-Adaptor | **0.3258** | **0.7255** | **0.5501** | **0.1657** | **0.4416** | **0.6986** |
| Architectural modifications | | | | | | |
| Without skip connection | 0.3243 | 0.6465 | 0.5110 | 0.1579 | 0.4133 | 0.6380 |
| With separate adapters | 0.3047 | 0.5488 | 0.3659 | 0.1463 | 0.3977 | 0.6148 |
| Regularization | | | | | | |
| Without prediction loss | 0.3235 | 0.6501 | 0.5456 | 0.1642 | 0.4078 | 0.6177 |
| Without reconstruction loss | 0.3245 | 0.6491 | 0.5439 | 0.1637 | 0.4127 | 0.6551 |
| Alternative losses | | | | | | |
| Sigmoid cross entropy | 0.3026 | 0.5917 | 0.4912 | 0.1567 | 0.4052 | 0.6702 |
| Contrastive loss (27) | 0.3046 | 0.5316 | 0.4822 | 0.1449 | 0.4091 | 0.6723 |
| Softmax cross entropy (28) | 0.3097 | 0.5452 | 0.4874 | 0.1346 | 0.4121 | 0.6549 |
| RankNet loss (29) | 0.3119 | 0.5511 | 0.4699 | 0.1599 | 0.4155 | 0.6428 |

Table 5: Ablation studies with variants of Search-Adaptor. We first modify regularizers and architectures of the original Search-Adaptor. Then, we only replace the proposed ranking loss with alternative ranking losses (30).

## 7 CONCLUSIONS

Pre-trained LLMs have shown great potential in a variety of downstream tasks. In this paper, we focus on pushing the capabilities of LLMs for information retrieval and search. We propose a canonical efficient adaptation method, Search-Adaptor, that can also be applied to LLMs available only via APIs. We demonstrate that Search-Adaptor significantly and consistently improves retrieval performance across diverse regimes of training data size, encoder type, and corpus set. Important future directions include generalizing the adaptation method to include partial tuning of the embedding models, as well as extensions to multi-modal data.

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

# A    DATA STATISTICS

## A.1    BEIR DATASETS

| Datasets | The number of train pairs | The number of test pairs | The number of corpus |
|---|---|---|---|
| NFCorpus | 110575 | 12334 | 3.6K |
| SciFact | 919 | 339 | 5K |
| Arguana | 703 | 703 | 8.67K |
| SciDocs | 14972 | 14956 | 25K |
| FiQA | 14166 | 1706 | 57K |
| Trec-Covid | 35460 | 30876 | 171K |
| Webis Touche 2020 | 1077 | 1137 | 382K |
| Quora | 7626 | 15675 | 523K |
| NQ | 2097 | 2104 | 2.68M |
| DBPedia | 5673 | 43515 | 4.63M |
| HotPotQA | 170000 | 14810 | 5.23M |
| Fever | 140085 | 7937 | 5.42M |
| Climate-fever | 2299 | 2382 | 5.42M |
| MSMarco | 532751 | 9260 | 8.84M |

Table 6: The statistics of the BEIR datasets (sorted by the number of corpus).

## A.2    MIRACL DATASETS

| Datasets | The number of train pairs | The number of test pairs | The number of corpus |
|---|---|---|---|
| Yoruba (yo) | 959 | 229 | 49043 |
| Swahilli (sw) | 9359 | 5092 | 131924 |
| Bengali (bn) | 16754 | 4206 | 297265 |
| Hindi (hi) | 11668 | 3494 | 506264 |
| Telugu (te) | 18608 | 1606 | 518079 |
| Thai (th) | 21293 | 7573 | 542166 |
| Indonesian (id) | 41358 | 9668 | 1446315 |
| Korean (ko) | 12767 | 3057 | 1486752 |
| Finnish (fi) | 20350 | 12008 | 1883509 |
| Arabic (ar) | 25382 | 29197 | 2061414 |
| Persian (fa) | 21844 | 6571 | 2207172 |
| Chinese (zh) | 13113 | 3928 | 4934368 |
| Japanese (ja) | 34387 | 8354 | 6953614 |
| Russian (ru) | 33921 | 13100 | 9543918 |
| Spanish (es) | 21531 | 6443 | 10373953 |
| French (fr) | 11426 | 3429 | 14636953 |
| Germany (de) | 2526 | 628 | 15866222 |

Table 7: The statistics of the MIRACL datasets (sorted by the number of corpus).

# B    METRICS

For tasks that involve retrieving information, normalized discounted cumulative gain (nDCG) (31) is a standard metric for evaluating performance. To define nDCG, we first consider discounted cumulative gain (DCG):

$$DCG(y, s) = \sum_i \frac{2^{y_i}}{\log_2(\text{rank}(s_i) + 1)},$$

where $s$ is the relevance score computed by the model and $y$ is the ground truth label. nDCG is then defined as $nDCG(y,s) = \frac{DCG(y,s)}{DCG(y,y)}$, where the denominator assumes the optimal case where the ranking of the scores ($s$) are exactly the same as the ranking of the ground truth label ($y$). nDCG@k is a widely used variation of nDCG where only the top $k$ scores are considered. In this paper, we use nDCG@10 as our main retrieval metric.

## C   HYPER-PARAMETERS

We summarize the hyper-parameters used to train Search-Adaptor. In all experiments, we utilized the fixed hyper-parameters that enable to apply Search-Adaptor without extensive hyper-parameter tuning.

| Hyper-parameters | Fixed values |
|---|---|
| Recovery loss coefficient ($\alpha$) | $\{0.0, 0.1, 1.0\}$ |
| Prediction loss coefficient ($\beta$) | $\{0.0, 0.01, 0.1\}$ |
| Batch size for training | 128 |
| Maximum number of training iterations | 2000 |
| Patience for early stopping | 125 |
| Learning rates | 0.001 |
| Optimizer | Adam |
| Negative pair subsampling ratio (compared with positive pairs) | 10 |

Table 8: Hyper-parameters used to train Search-Adaptor in all experiments.

## D   ADDITIONAL EXPERIMENTS

We include the additional results of Search-Adaptor with GTR-Large[2] (32) as the base embedding models. As can be seen in Table. 9, the results are consistent with the above results that Search-Adaptor shows consistent and significant improvements on top of the GTR-Large model.

| Datasets | GTR-Large Model | | |
|---|---|---|---|
| | Zero-shot | Search-Adaptor | Gains (%) |
| NFCorpus | 0.3148 | **0.3242** | 2.99% |
| SciFact | 0.5331 | **0.7469** | 40.11% |
| Arguana | 0.5139 | **0.6360** | 23.76% |
| SciDocs | 0.1657 | **0.1687** | 1.81% |
| FiQA | 0.4069 | **0.4265** | 4.82% |
| Trec-Covid | 0.6912 | **0.7481** | 8.23% |
| Webis Touche 2020 | 0.2723 | **0.3227** | 18.51% |
| Quora | 0.8428 | **0.8795** | 4.35% |
| Average | 0.4676 | **0.5315** | 13.68% |

Table 9: Performance improvements with Search-Adaptor on top of GTR-Large embedding model.

## E   QUALITATIVE ANALYSIS

First, we compute the cosine similarity between query and corpus, before and after Search-Adaptor training. Then, we plot the cosine similarity between relevant / irrelevant query corpus pairs.

As can be seen in Fig. 3, after Search-Adaptor training, the distribution differences between relevant and irrelevant pairs' cosine similarity are larger which means that we can identify the relevant corpus per each query better.

---

[2]https://huggingface.co/sentence-transformers/gtr-t5-large

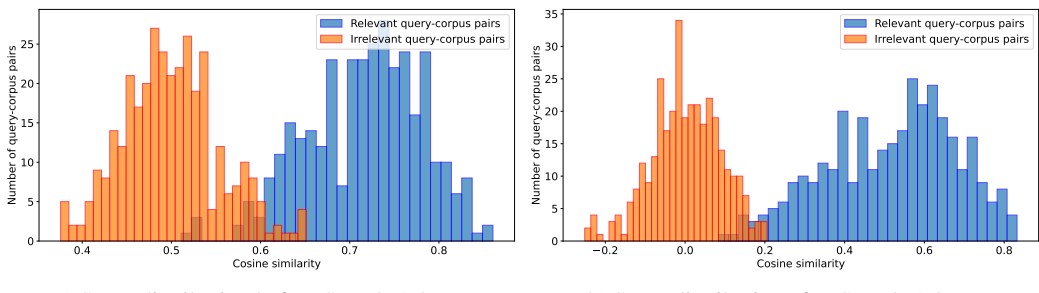

(a) Score distribution before Search-Adaptor      (b) Score distribution after Search-Adaptor

Figure 3: Cosine similarity score distributions before and after Search-Adaptor.

To further understand the distribution difference of query / corpus embeddings before and after Search-Adaptor training, we plot tSNE graphs of query and corpus embeddings.

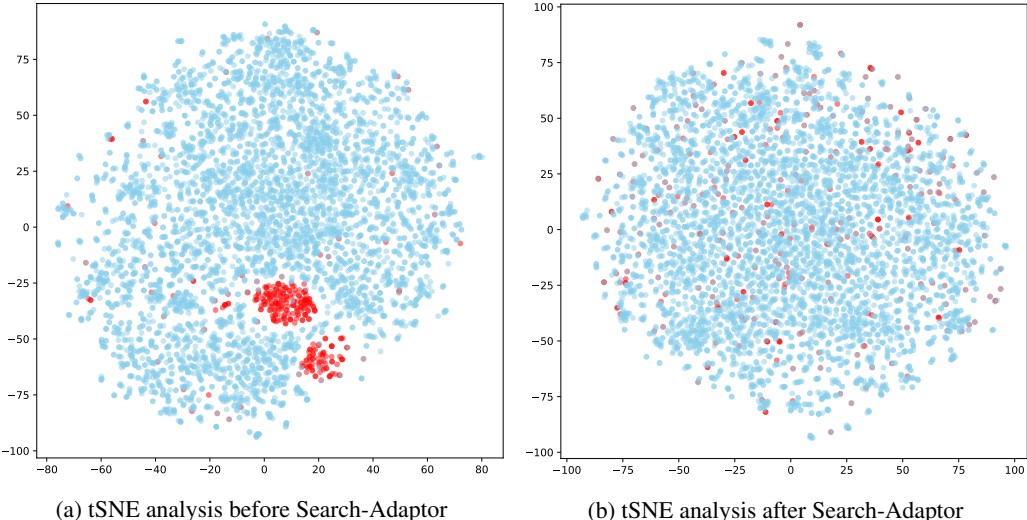

(a) tSNE analysis before Search-Adaptor      (b) tSNE analysis after Search-Adaptor

Figure 4: tSNE distributions before and after Search-Adaptor. Red represents query embedding and blue represents corpus embedding.

Fig. 4 shows the impact of Search-Adaptor. The left figure shows that the original query and corpus embeddings are quite distinct. Most query embeddings are located in the restricted region. On the other hand, after training with Search-Adaptor, query embedding distribution is observed to better overlap with the corpus embedding distribution, which could result in more robust retrieval.

We further investigate the success and failure cases of Search-Adaptor in comparison to the zero-shot baseline. Bold represents the relevant corpus to the query.

As can be seen in Table. 10 and 11, in failure cases, Search-Adaptor still can retrieve the relevant corpus in the top-3 corpus but the ranking is lower than the baseline. For the success cases, Search-Adaptor can retrieve the correct corpus even though the baseline is completely failed. Quantitatively, with 300 test samples, there are 9 cases where Search-Adaptor can retrieve the correct corpus in top-3 but Baseline cannot retrieve any correct corpus in top-3. But there is no case for the opposite.

| Query | Baseline Retrieval | Search-Adaptor Retrieval |
|---|---|---|
| Suboptimal nutrition is not predictive of chronic disease | Maternal and child undernutrition: consequences for adult health and human capital | **Global, regional, and national comparative risk assessment of 79 behavioural, environmental and occupational, and metabolic risks or clusters of risks, 1990–2015: a systematic analysis for the Global Burden of Disease Study 2015** |
| | Effect of women's nutrition before and during early pregnancy on maternal and infant outcomes: a systematic review. | Dietary quality among men and women in 187 countries in 1990 and 2010: a systematic assessment |
| | Dietary quality among men and women in 187 countries in 1990 and 2010: a systematic assessment | Biomarkers of endothelial dysfunction and risk of type 2 diabetes mellitus. |
| The PRR MDA5 is a sensor of RNA virus infection. | Ribose 2-O-methylation provides a molecular signature for the distinction of self and non-self mRNA dependent on the RNA sensor Mda5 | **Immune signaling by RIG-I-like receptors.** |
| | **Immune signaling by RIG-I-like receptors.** | Ribose 2-O-methylation provides a molecular signature for the distinction of self and non-self mRNA dependent on the RNA sensor Mda5 |
| | RIG-I-mediated antiviral responses to single-stranded RNA bearing 5'-phosphates. | RIG-I-mediated antiviral responses to single-stranded RNA bearing 5'-phosphates. |
| A deficiency of vitamin B12 increases blood levels of homocysteine. | Preventing coronary heart disease: B vitamins and homocysteine. | **Folic acid improves endothelial function in coronary artery disease via mechanisms largely independent of homocysteine lowering.** |
| | Effect of homocysteine lowering on mortality and vascular disease in advanced chronic kidney disease and end-stage renal disease: a randomized controlled trial. | **Randomized trial of folic acid supplementation and serum homocysteine levels.** |
| | Hyperhomocysteinemia and atherosclerotic vascular disease: pathophysiology, screening, and treatment. off. | The effect of folic acid supplementation on plasma homocysteine in an elderly population. |

Table 10: Success cases: Examples of query and top-3 retrieved documents where relevant documents are ranked higher in Search-Adaptor in comparison to baseline. Top-3 retrieved documents' titles are listed above.

| Query | Baseline Retrieval | Search-Adaptor Retrieval |
|---|---|---|
| Antibiotic induced alterations in the gut microbiome reduce resistance against Clostridium difficile | **Antibiotic-induced shifts in the mouse gut microbiome and metabolome increase susceptibility to Clostridium difficile infection** | Precision microbiome reconstitution restores bile acid mediated resistance to Clostridium difficile |
| | Precision microbiome reconstitution restores bile acid mediated resistance to Clostridium difficile | **Antibiotic-induced shifts in the mouse gut microbiome and metabolome increase susceptibility to Clostridium difficile infection** |
| | Role of gut commensal microflora in the development of experimental autoimmune encephalomyelitis. | Microbiome-driven allergic lung inflammation is ameliorated by short-chain fatty acids |
| The genomic aberrations found in matasteses are very similar to those found in the primary tumor. | **Evolution of metastasis revealed by mutational landscapes of chemically induced skin cancers** | Intratumor heterogeneity and branched evolution revealed by multiregion sequencing. |
| | Molecular characterization of endometrial cancer: a correlative study assessing microsatellite instability, MLH1 hypermethylation, DNA mismatch repair protein expression, and PTEN, PIK3CA, KRAS, and BRAF mutation analysis. | Diverse tumorigenic pathways in ovarian serous carcinoma. |
| | Deregulated DNA polymerase beta induces chromosome instability and tumorigenesis. | **Evolution of metastasis revealed by mutational landscapes of chemically induced skin cancers** |
| Incidence rates of cervical cancer have increased due to nationwide screening programs based primarily on cytology to detect uterine cervical cancer. | **Mass screening programmes and trends in cervical cancer in Finland and the Netherlands.** | The effect of mass screening on incidence and mortality of squamous and adenocarcinoma of cervix uteri. |
| | The effect of mass screening on incidence and mortality of squamous and adenocarcinoma of cervix uteri. | **Mass screening programmes and trends in cervical cancer in Finland and the Netherlands.** |
| | Efficacy of human papillomavirus testing for the detection of invasive cervical cancers and cervical intraepithelial neoplasia: a randomised controlled trial. | Efficacy of human papillomavirus testing for the detection of invasive cervical cancers and cervical intraepithelial neoplasia: a randomised controlled trial. |

Table 11: Failure cases: Examples of query and top-3 retrieved documents where relevant documents are ranked higher in baseline in comparison to Search-Adaptor. Top-3 retrieved documents' titles are listed above.

