# OpenReview forum: "Search-Adaptor: Text Embedding Customization for Information Retrieval"
_ICLR.cc/2024/Conference — Submitted to ICLR 2024_

### Official Review · Reviewer_W3Mq · 2023-10-20

**Soundness:** 3 good
**Presentation:** 2 fair
**Contribution:** 2 fair
**Rating:** 3
**Confidence:** 4

**Summary:**

The paper discusses the utilization of pre-trained Large Language Models (LLMs) for enhancing information retrieval and search. The authors proposes approach called Search-Adaptor, aimed at efficiently and robustly customizing LLMs for information retrieval. Search-Adaptor modifies the text embeddings from pre-trained LLMs and can be seamlessly integrated with various LLMs, including those accessible via APIs. This method is a lightweight method, which shows its effectiveness in zero shot retrieval tasks.

**Strengths:**

1. The proposed method is simple but effectiveness.
2. The method can be directly used to learn the query representations for retrieval.

**Weaknesses:**

1. The motivation is unclear. The representation is indeed a critical topic that lots of researchers have paid attention to. Nevertheless, the main reason is that the queries usually contain less information than documents. Previous work usually employs query expansion methods to enhance the query representations, such as using KG, PRF, and LLMs. The other possible motivation is that the idea can adapt the LLMs to utilize the retrieval tools. Overall, I suggest the authors should carefully describe their motivations.
2. Widely used retrieval benchmarks, such as MSMARCO is not used in experiments.
3. The experimental results are not good on some QA tasks, such as HotpotQA and NQ.
4. No query extension baselines are compared. The Promptagator and PRF models should be compared. Experimental results on coCondenser or GTR should also be conducted.
5. The differences between this work and others[1-2] should be described.
6. The citation formulation seems wrong.

[1] Precise Zero-Shot Dense Retrieval without Relevance Labels.
[2] Promptagator: Few-shot dense retrieval from 8 examples.

**Questions:**

Why do you not choose some existing dense retrieval models for evaluation.

---

> ### Author Response · Authors · 2023-11-23
> **Responses to the reviewer W3Mq (1/2)**
>
> Thanks for all your valuable feedback that has helped us to improve our manuscript. Based on your suggestions, we have included various additional experiments in the revised manuscript. Please see the detailed responses below and let us know if you have any further questions or comments.
>
> **Answer 1**: The main objective of Search-Adaptor is modifying the embeddings such that relevant query and corpus embeddings are closer and irrelevant query and corpus embeddings are farther. In that case, query expansion approaches and Search-Adaptor are not incompatible. We agreed that queries usually include less information than the corpus. It is possible to first utilize query expansion to modify the query; then, we apply Search-Adaptor on top of expanded queries and corpus.
>
> **Answer 2:** Thanks for the suggestion. We have added the results using MSMARCO with Google gecko@latest encoder model. Note that we use the train data for training, dev data for evaluation in the original MSMARCO data (this is standard on MSMARCO experiments). In terms of NDCG@10, baseline (Google gecko@latest) achieves 0.2922 and Search-Adaptor achieves 0.3177 which is 8.73% improvement.
>
> **Answer 3:** Results in the original manuscript fixed the critical coefficients as constant (alpha = 0.1, beta = 0.01) for the simplicity. For the robust and consistent improvements, we add a simple hyper-parameter tuning as selecting the model with the best dev nDCG@10 performance among the combinations of alpha in [0.0, 0.1, 1.0] and beta in [0.0, 0.01, 0.1] (total 9 combinations). With this simple hyper-parameter tuning, we can achieve consistent improvements across the entire datasets. All the results are updated in the revised manuscript (see Table 1 to 5). Note that the hyper-parameter tuning can be parallelized to mitigate impact on the computation time.
>
> **Answer 4:** Actually, Promptagator and Query expansions are not the competitor alternatives to Search-Adaptor, but rather approaches that can complement Search-Adaptor. First, we can apply query expansion on top of the original given query-corpus pairs to enhance the information in the query side. Then, we can apply Search-Adaptor on top of expanded query and corpus pairs. These can be combined into two approaches to yield superior retrieval systems. Note that query expansion methods including (PRF) do not utilize the given query-corpus pairs when expanding queries.
>
> We also include the additional results of Search-Adaptor with GTR-Large (for coConderser, we will add the results in the camera-ready) as the base embedding models. As can be seen in Table 9, the results are consistent with other results in the manuscript that Search-Adaptor shows consistent and significant improvements on top of the GTR-Large model.

---

> ### Author Response · Authors · 2023-11-23
> **Responses to the reviewer W3Mq (2/2)**
>
> **Answer 5**: Given a query, HyDE [1] first utilizes the Instruct-GPT to generate hypothetical documents. Then, it converts the hypothetical document into the embedding space and finds the most similar corpus in the embedding space. The main objective of HyDE is quite different. Search-Adaptor utilizes “given” query-corpus pairs to improve the retrieval performance. On the other hand, HyDE does not utilize any labeled query-corpus pairs. Given the query, it first generates pseudo-corpus and then finds the similar real corpus in the embedding space. The setting in Search-Adaptor is “supervised” while for HyDE, the setting is unsupervised. Another big limitation of HyDE is that at inference, it has the cost/latency coming from one LLM (e.g., Instruct-GPT) inference and one encoder inference. Usually, one LLM inference takes much longer than one encoder inference (due to the decoding). Therefore, the cost/latency of HyDE would be much higher than Search-Adaptor, an aspect critical in retrieval systems.
>
> Promptagator is proposed for the few-shot (<10 samples) regime. First, using few-shot samples as the prompt, Promptagator generates pseudo-queries that are relevant to the corpus. Then, Promptagator trains the round-trip filtering method to filter out some noisy pseudo-query corpus pairs. Lastly, using the few shot query-corpus pairs and filtered out pseudo-query corpus pairs, it trains the contriever model for the final retrieval task. Search-Adaptor and Promptagator have clear differences. First, the setting - Promptagator is in the few shot setting whereas Search-Adaptor is in the supervised learning setting. Second, Promptagator has not been demonstrated with an API-based encoder. It may need some methodological modification for round-trip filter training and downstream retrieval task fine-tuning. Lastly, the generated pseudo-query corpus pairs can be also utilized to train Search-Adaptor. We actually try to train Search-Adaptor using filtered pseudo-query corpus pairs in addition to the given original query corpus pairs. Unfortunately, if we have sufficient data (>100 samples), the performance might even degrade with the pseudo-query corpus pairs as the generated queries can be quite different from the original query. Also, the accuracy of retrieving the correct documents for pseudo-query is much higher than the accuracy of retrieving the correct documents for original queries. This implies that the generated queries by the Promptagator would be too straightforward.
>
> **Answer 6:** We used the citation format of IEEE Transaction which is acceptable for ICLR publication.
>
> **Answer 7:** Note that Search-Adaptor can be applicable on top of any embedding model (regardless of parameter access type). Any dense retrieval model, based on their embeddings used for retrieval, can benefit from Search-Adaptor. To further address the comment, we have included additional experiments of Search-Adaptor applying on top of another dense retrieval model (GTR-Large). Please see the results in Answer 4.

---

### Official Review · Reviewer_ftHT · 2023-10-28

**Soundness:** 2 fair
**Presentation:** 2 fair
**Contribution:** 2 fair
**Rating:** 5
**Confidence:** 5

**Summary:**

This paper proposes amethod, Search-Adaptor, to customize LLMs for information retrieval.
Although multiple parameter-efficient fine-tuning methods reduce the risks of overfitting and provides computational gains,
they need full access to the LLM’s parameters to fine-tune the model, which may not be possible with API-based LLMs.
Search-Adaptor places a small adapter network on top of fixed LLMs to modify the original text embedding generated by pre-trained LLMs, and can be integrated with any LLM, including those only available via APIs.

**Strengths:**

**The architecture of Search-Adaptor**
- This paper proposes a novel adaptation framework, Search-Adaptor, places a small adapter network on top of fixed LLMs to modify the pre-trained text embeddings.
- This architecture does not require access to the parameters of the pre-trained LLMs only the inference outputs of the model are needed,
and can also be applied to LLMs available only via APIs, while multiple parameter-efficient fine-tuning methods need full access to the LLM’s parameters to fine-tune the model, which may not be possible with API-based LLMs.

**Components of Search-Adaptor to  retrieval performance even with the small data regime**
- To customize embeddings extracted from the pre-trained LLMs, authors introduce a novel ranking loss that is directly utilized for Search-Adaptor training.
- To avoid forgetting too much information from the pre-trained LLMs, authors design two regularization methods Recovery and Prediction.

**Weaknesses:**

**Compatibility with Goals**
- It is not clear whether the customization perspective is domain or individual. This makes the approach more detailed.
- The adapter-based approach is promising, but its strengths relative to other approaches, such Prompt tuning, Chain-of-Thought (CoT), Tree of Thoughts (ToT), and, Retrieval Augmented Generation (RAG)  have not been described.

**Experimental settings and results**
- The results may vary depending on the size and quality of the data used for customization and its compatibility with LLM, but this has not been fully examined.
- No qualitative assessment of performance or Search-Adaptor strengths and weaknesses were stated.
- Due to the lack of explanation of the values and interpretation of the Table 5, the validity of the authors' claims can not be determined.

**Questions:**

- Are there any changes to the tokenizer?
- See Weaknesses

---

> ### Author Response · Authors · 2023-11-23
> **Responses to the reviewer ftHT (1/2)**
>
> Thanks for recognizing the novelty and usefulness of our methodology. Please see the detailed responses below and let us know if you have any further questions or comments.
>
> **Answer 1:** In the experiments, we train different Search-Adaptor models per each dataset (individual approach). For instance, in Table 2, we use different Search-Adaptor models per each language. Note that the training cost of Search-Adaptor is marginal and it can be effective with a small amount of data. In addition, during serving, we just need a separate adaptor on top of shared embedding APIs. These make Search-Adaptor quite practical when applying individually.
>
> Note that it is possible to apply customization per domain as well, so that larger and more diverse tuning data can be employed. However, since Search-Adaptor is efficient to train and serve, it might not be preferred.
>
> **Answer 2:** Thanks for suggesting various additional baselines. We will add new baselines (LoRA, prompt tuning) in the camera-ready. Note that both prompt tuning and LoRA need full access to the model; thus, we can only integrate with publicly available open-source embedding models (e.g., Sentence-T5 Base).
>
> Chain-of-Thought (CoT) and Tree of Thoughts (ToT) are not proposed to tune with labeled data samples, but to improve the reasoning of the LLMs. Furthermore, due to the limited prompt token size, they cannot utilize the entire query-corpus pairs. Regarding the Retrieval Augmented Generation (RAG), better embeddings (e.g., with Search-Adaptor), can help better retrieval in RAG which results in better answer generation. Therefore, we think CoT, ToT and RAG are not the direct baselines of Search-Adaptor.
>
> **Answer 3:** Results in the original manuscript fixed the critical coefficients as constant (alpha = 0.1, beta = 0.01) for the simplicity. For the robust and consistent improvements, we add a simple hyper-parameter tuning as selecting the model with the best dev nDCG@10 performance among the combinations of alpha in [0.0, 0.1, 1.0] and beta in [0.0, 0.01, 0.1] (total 9 combinations). With this simple hyper-parameter tuning, we can achieve consistent improvements across the entire datasets. All the results are updated in the revised manuscript (see Table 1 to 5). Note that the hyper-parameter tuning can be parallelized to mitigate impact on the computation time.
>
> **Answer 4:** We appreciate your suggestions on qualitative analyses. In the revised manuscript, we add multiple qualitative analyses to understand Search-Adaptor better (see Appendix E).
>
> First, we compute the cosine similarity between query and corpus, before and after Search-Adaptor training. Then, we plot the cosine similarity between relevant / irrelevant query corpus pairs.
>
> As can be seen in Fig. 3, after Search-Adaptor training, the distribution differences between relevant and irrelevant pairs’ cosine similarity are larger which means that we can identify the relevant corpus per each query better.
>
> To further understand the distribution difference of query / corpus embeddings before and after Search-Adaptor training, we plot tSNE graphs of query and corpus embeddings.
>
> Fig. 4 shows the impact of Search-Adaptor. The left figure shows that the original query and corpus embeddings are quite distinct. Most query embeddings are located in the restricted region. On the other hand, after training with Search-Adaptor, query embedding distribution is observed to better overlap with the corpus embedding distribution, which could result in more robust retrieval.
>
> We further investigate the success and failure cases of Search-Adaptor in comparison to the zero-shot baseline (Table 10 and 11). Bold represents the relevant corpus to the query.
>
> As can be seen in the examples (in Table 10 and 11), in failure cases, Search-Adaptor still can retrieve the relevant corpus in the top-3 corpus but the ranking is lower than the baseline. For the success cases, Search-Adaptor can retrieve the correct corpus even though the baseline is completely failed. Quantitatively, with 300 test samples, there are 9 cases where Search-Adaptor can retrieve the correct corpus in top-3 but Baseline cannot retrieve any correct corpus in top-3. But there is no case for the opposite.

---

> ### Author Response · Authors · 2023-11-23
> **Responses to the reviewer ftHT (2/2)**
>
> **Answer 5**: In Table 5, we would like to understand the source of gains in the Search-Adaptor approach. So, we make various modifications to the Search-Adaptor: (i) different architecture, (ii) different regularization, (iii) different losses. First, using different losses makes the biggest performance degradation which represents the importance of our ranking loss function. In addition, if we use separate adapters for query and corpus, it also makes a noticeable performance drop. This shows the importance of “shared embedding space” between query and corpus for the retrieval application. Lastly, skip connection, two regularization functions also bring additional performance gains but the impact is lower than our ranking losses. We have clarified all these in the revised version of the manuscript.
>
> **Answer 6:** We do not change the tokenizer of the embedding APIs.

---

### Official Review · Reviewer_VfLX · 2023-11-01

**Soundness:** 4 excellent
**Presentation:** 3 good
**Contribution:** 4 excellent
**Rating:** 8
**Confidence:** 3

**Summary:**

Search-Adaptor modifies the original text embedding
generated by pre-trained LLMs, and can be integrated with any LLM, including
those only available via APIs.  Impressive gains are achieved on a number of benchmarks.

**Strengths:**

The proposed method is simple and effective.  It is similar to learning to rank, but updated to use more modern methods.  This paper is likely to be highly cited.

**Weaknesses:**

The benchmarks in the evaluation are claimed to be "real-world," but no evidence is provided for that claim, and it is almost surely incorrect.  Let us not confuse toy benchmarks with real tasks.

The discussion of section 4.1 would be easier going if the notation were clearly defined before introducing the ranking objective. y is defined above the equation, and s is defined well below the equation.  I should not have to hunt around for simple things like this.  Please do not use notation before it is defined.  E is also used before it is defined.

It is a shame that there isn't a github showing how to replicate the experiments.

Many of the gains in Table 1 are positive, but some are not.

**Questions:**

How much computation went into these experiments?

Since some of the gains in Table 1 are positive and some are not, would it be worthwhile to ensemble zero-shot and search-adaptor?

---

> ### Author Response · Authors · 2023-11-23
> **Responses to the Reviewer VfLX**
>
> Thanks for all your valuable feedback and positive comments on our manuscript. Please see the detailed responses below and let us know if you have any further questions or comments.
>
> **Answer 1:** We have removed the term “real-world” in the revised manuscript when describing the datasets that we used in the experiments. Note that these benchmarks reflect various use cases from social media, healthcare, and knowledge retrieval systems, however, as the reviewer noted, their sizes or the complexities of the queries might have certain limitations when all real-world use cases are considered.
>
> **Answer 2:** We agree with the reviewer’s comment and reorganized Section 4, switching the order of “Adapting Fixed LLMs” and “Ranking Objective” to make sure all the notations are described before introducing them in the equations. Now, all the notations (e.g., E, y, s) are described before being introduced.
>
> **Answer 3:** We have a plan to publish the codebase upon acceptance.
>
> **Answer 4:** Results in the original manuscript fixed the critical coefficients as constant (alpha = 0.1, beta = 0.01) for the simplicity. For the robust and consistent improvements, we add a simple hyper-parameter tuning as selecting the model with the best dev nDCG@10 performance among the combinations of alpha in [0.0, 0.1, 1.0] and beta in [0.0, 0.01, 0.1] (total 9 combinations). With this simple hyper-parameter tuning, we can achieve consistent improvements across the entire datasets. All the results are updated in the revised manuscript (see Table 1 to 5). Note that the hyper-parameter tuning can be parallelized to mitigate impact on the computation time.
>
> **Answer 5:** Majority of the computation is based on text embedding extraction which often depends on the query-per-second (QPS) limits of text embedding APIs (in most cases, 8 NVIDIA V100 GPUs are used for the inference, https://arxiv.org/pdf/2308.03281.pdf). After extracting all the embeddings, training Search-Adaptor takes less than two hours with any dataset (including large datasets such as HotPotQA, Fever, French) using 8 NVIDIA V100 GPUs, without any specially-optimized implementation.
>
> **Answer 6:** Based on better hyper-parameter tuning (using model selection by dev nDCG@10), we can achieve positive improvements for all datasets. Please see more details in Answer 4.
> The ensemble idea is quite interesting - we can even ensemble across multiple different adaptors trained with different random seeds in addition to the zero-shot case. We leave this important exploration to future work.

---

### Meta-Review · Area_Chair_pKBU · 2023-12-06

**Metareview:**

The paper addresses LLM-based information retrieval and proposes a Search-Adaptor method that inserts a low-cost adaptor layer to learn a customized embedding of the retrieved text.

Reviewers raised certain concerns about the experimentation. In addition, my main concern is the paper does not provide strong technicality as the contribution is mainly about inserting an adaptor layer.

**Justification For Why Not Higher Score:**

Reviewers raised certain concerns about the experimentation. In addition, my main concern is the paper does not provide strong technicality as the contribution is mainly about inserting an adaptor layer.

**Justification For Why Not Lower Score:**

N/A

---

### Decision · Program_Chairs · 2024-01-16

Reject